# Cyanobacteria as Chassis for Industrial Biotechnology: Progress and Prospects

**DOI:** 10.3390/life6040042

**Published:** 2016-11-30

**Authors:** Lamya Al-Haj, Yuen Tin Lui, Raeid M.M. Abed, Mohamed A. Gomaa, Saul Purton

**Affiliations:** 1Biology Department, College of Science, Sultan Qaboos University, Al-Khoud, P.O. Box 36, Muscat 123, Oman; rabed@squ.edu.om (R.M.M.A.); moh.gomaa@live.com (M.A.G.); 2Institute of Structural & Molecular Biology, University College London, London WC1E 6BT, UK; y.lui.12@ucl.ac.uk (Y.T.L.); s.purton@ucl.ac.uk (S.P.)

**Keywords:** cyanobacteria, biotechnology, biofuels, bio-products, genetic engineering

## Abstract

Cyanobacteria hold significant potential as industrial biotechnology (IB) platforms for the production of a wide variety of bio-products ranging from biofuels such as hydrogen, alcohols and isoprenoids, to high-value bioactive and recombinant proteins. Underpinning this technology, are the recent advances in cyanobacterial “omics” research, the development of improved genetic engineering tools for key species, and the emerging field of cyanobacterial synthetic biology. These approaches enabled the development of elaborate metabolic engineering programs aimed at creating designer strains tailored for different IB applications. In this review, we provide an overview of the current status of the fields of cyanobacterial omics and genetic engineering with specific focus on the current molecular tools and technologies that have been developed in the past five years. The paper concludes by giving insights on future commercial applications of cyanobacteria and highlights the challenges that need to be addressed in order to make cyanobacterial industrial biotechnology more feasible in the near future.

## 1. Introduction

Cyanobacteria represent an extremely diverse, yet highly specialised group of prokaryotic organisms found in a wide range of environmental habitats [1]. Cyanobacteria have been regarded as attractive laboratory models for genetic studies of fundamental processes such as carbon and nitrogen fixation [2,3]. This is mainly because they are amenable to genetic manipulation by the techniques already developed for the intensively studied heterotrophic bacteria such as *E. coli* [4]. The field of molecular biology for cyanobacteria has witnessed enormous development in the past two decades going from almost the absence of any knowledge in the mid-seventies to the development of transformation systems in the eighties [5,6]. In the nineties, this field turned into a full-fledged unique and promising field of research [7,8,9] due to the determination of the full sequence of the cyanobacterium *Synechocystis* PCC 6803 in 1996 [10]. Nowadays, the full genomic sequences of 128 cyanobacterial strains are available, hence making their genetic manipulation even more feasible [11].

Recently, a lot of attention has been directed towards the development of cyanobacteria as industrial biotechnology (IB) platforms by employing bioengineering and synthetic biology tools [12,13,14,15]. Cyanobacteria have great potential in the production of renewable fuels, various chemicals and nutritional products. A list of various drop-in biofuels that have been produced in engineered cyanobacteria are detailed in [16,17] and Table 1. The fact that engineered cyanobacteria have the potential to produce compounds that act as drop-in fuels that do not require additional refining or chemical modification for use in an engine [18], and that their composition can be tailored by genetic manipulation, make cyanobacterial-produced fuel molecules excellent options as future fungible fuels [6]. In a recent study, the combustion and emissions of terpenes in a diesel engine were examined with a view to their biological production in cyanobacteria [19]. The study highlighted the potential of genetically engineering the isoprenoid pathway of *Synechocystis* 6803 for the production of isoprenoid-derived fuels that can act as drop-in fuels in diesel engines [19].

Additionally, cyanobacteria are being optimised to overproduce numerous products of value [20]. For example, naturally occurring cyanobacterial phycobiliproteins are commonly used in fluorescent tagging applications in research [21] in conjugation with monoclonal and polyclonal antibodies to make fluorescent antibody reagents that are used for cell sorting [22]. Other natural products are used as food dyes or pigments in cosmetics [23]. Some cyanobacterial strains are used as vectors for the delivery of mosquitocidal toxins [21], while nitrogen-fixing strains are commonly used as natural fertilisers [23]. Chemical products such as fish growth hormones [21], polyhydroxyalkanoates (PHA’s) [22,24] and polyunsaturated fatty acids (PUFA) [25] have been produced from genetically engineered cyanobacteria. *Synechococcus* has been genetically engineered for the heterologous expression of a single gene (*efe*) from *Pseudomonas syringae* that led to the successful synthesis of ethylene [8], one of the most important building blocks in synthetic chemical industry [26]. Furthermore, acetone was recently produced by modified *Synechocystis* 6803 with a reported yield of 36.0 mg/L [27]. A summary of some of the various cyanobacterial products can be found in Figure 1, and [28].

Although synthetic biology tools have contributed significantly to unlocking cyanobacteria’s potential for these functions, these tools are still far behind those established for *S. cerevisiae* and *E. coli* [29]. Nevertheless, in recent years, there have been significant advances in cyanobacterial strain improvement using genetic engineering and synthetic biology [30]. It is worth mentioning that most of the efforts for developing genetic engineering tools have been aimed at model or heavily studied cyanobacterial strains such as the filamentous, nitrogen-fixing *Anabaena* sp. strain PCC 7120 [31,32,33], and the naturally competent *Synechococcus* 7942 [34,35,36,37], *Synechococcus* 7002 [38,39,40,41,42,43,44] and *Synechocystis* 6803 [16,45,46,47,48,49,50,51,52]. This review discusses the recent developments in the field of cyanobacterial genetic modification in the past five years and how these tools can be employed to increase the efficiency of cyanobacteria in industrial biotechnology applications.

## 2. Cyanobacterial “Omics”

The model organism *Synechocystis* sp. PCC 6803 (hereafter *Synechocystis* 6803) in 1996 was the first photosynthetic organism to have its genome sequenced [53]. Advances in sequencing technologies and the decrease in their costs have led to more and more strains of cyanobacteria being fully sequenced [54,55]. To date, CyanoBase (http://genome.microbedb.jp/cyanobase), one of the most widely used databases for cyanobacterial strains, contains 39 completed cyanobacterial genome sequences while the National Centre for Biotechnological Information (NCBI) (http://www.ncbi.nlm.nih.gov) contains 111 sequenced strains. The availability of these genomes has a huge impact on the directed molecular biology work that can be carried out. Genomic sequences can be used to identify pathways useful for the production of biotechnologically relevant compounds. For example, a comparison between the genomic sequences of alkane-producing and non-producing cyanobacteria has led to the identification of the alkane-producing pathway [56].

Microarray data measures the level of expression of a large number of genes within a genome. There is a large amount of microarray data for *Synechocystis* 6803 looking at levels of gene expression in different conditions that can be found in the databases Gene Expression Omnibus (GEO) [57], ArrayExpress [58] and Kyoto Encyclopedia of Genes and Genomes (KEGG) [59]. Although a lot of microarray data is on the two main model species, *Synechocystis* 6803 and *Synechococcus elongatus* PCC7942, there is also data on other cyanobacteria on the GEO database such as *Prochlorococcus marinus* MED4 and *Nostoc punctiforme* PCC73102. Recently, RNA-sequencing has started being used to study the transcriptome of cyanobacteria. For instance, RNA-sequencing transcriptomic analysis of the diazotrophic *Anabaena* sp. strain 90 revealed the upregulation of phosphorous transport and assimilation genes (*pho* regulon) during phosphorous deprivation in algal blooms [60]. RNA-sequencing was used to identify genes in *Synechocystis* 6803 that could be engineered to make ethanol-resistant strains [61]. In another study, RNA-sequencing transcriptomic analysis revealed a gene encoding a phosphatase protein (*slr1860*) that promoted ethanol tolerance in *Synechocystis* 6803 [62].

Metabolomics, transcriptomics and proteomics have been used to improve biofuel tolerances in *Synechocystis* 6803, which provided valuable correlations between genes, protein expression and production titers of different biotechnological products [63,64]. In a study on isoprene production from *Synechocystis* 6803, the metabolomic and transcriptomic profiles were analysed under metabolic stress [64]. The metabolomic profile suggested that the limiting factor of isoprene production under stress might be due to an insufficient precursor level, while the transcriptome revealed an upregulation of RNA characteristic of acclimation to stress [64]. Tolerance of *Synechocystis* 6803 towards other products, like butanol, was investigated using transcriptomics and it was revealed that the gene encoding HspA, a small heat shock protein, was induced when *Synechocystis* 6803 was grown in the presence of butanol. By overexpressing this gene, *Synechocystis* was more tolerant to butanol [65]. Proteomics analysis identified genes in *Synechocystis* 6803 that are induced in response to hexane, butanol and ethanol [66,67,68]. The information gained can be used to engineer tolerance into *Synechocytsis* 6803 in the hope of optimizing the production of biofuels and other toxic products in this model organism. Such studies can later be expanded to include other cyanobacterial strains.

Over the past decade, *Synechocystis* 6803 metabolism has been modeled [69,70,71,72,73,74,75,76,77,78,79] to understand and improve cyanobacterial biofuel production. In 2009, the first genome-scale model of *Synechocystis* 6803 to predict the effect of biofuel production was produced [80]. The in silico predictions introducing an ethanol-producing pathway into *Synechocystis* 6803 matched the experimental data [80]. This represents a good example of how transcriptomics and proteomics can assist in a more directed research approach that can immensely help in saving time. Recently, a visual representation of the metabolic network of *Synechocystis* 6803 called FAME was produced (http://f-a-m-e.org/synechocystis/), which can incorporate experimental and computational data. FAME can be used as an aid to engineer the metabolic pathway of *Synechocystis* 6803 for the production of a range of high value products and biofuels [81].

Thus far, the majority of the transcriptomic, proteomic, metabolomic and systems biology data available has focused on *Synechocystis* 6803 with little work performed on other cyanobacterial strains. Although a genome-scale model has been produced for *Synechoccocus* 7942, [82] modeling the nitrogen-fixing *Cyanothece* has already been attempted [83,84]. However, for cyanobacterial strains without a well-annotated genome, producing a genome-scale model will be challenging.

## 3. Classical Genetic Engineering Technology

### 3.1. Cyanobacterial Transformation

Transformation of cyanobacteria has been thoroughly reviewed [4,22,85,86] and the methodology has been well described [5,69]. The ability to transform cyanobacteria has been the drive for the development of the molecular biology of these organisms and has been the basis of many of their biotechnological applications [70]. Several cyanobacterial strains can be transformed with exogenously added DNA [4] or can accept mobilised plasmid DNA from *E. coli* by conjugation [15]. Natural transformation was first described for *Synechococcus* sp. strain PCC7942 [85] and hence is considered the primary means for gene transfer in unicellular cyanobacteria [6]. *Synechococcus* and *Synechocystis* species are among the few species that are known to be naturally transformable by exogenous DNA with little known about why natural transformation has thus far been restricted to certain unicellular cyanobacteria. This may be due to the presence of extracellular nucleases found in heterocyst-forming filamentous cyanobacteria that results in the degradation of the incoming DNA [71]. The natural transformable nature of *Synechococcus* and *Synechocystis* species explains why a lot of focus has been directed to the genetic engineering of these species. There are many factors that affect the frequency and efficiency of transformation in cyanobacteria, which can vary between 10^−3^ and 10^−8^ [4]. These include the used strain, the length, form and concentration of used DNA and also the method used for transformation [72]. The optimum conditions for transformation of the model organism *Synechocystis* sp. PCC 6803 are well described [72,73].

Since the discovery of cyanobacterial transformation in the 1970s, many techniques such as electroporation and ultrasonic treatment have been developed and applied to several cyanobacterial strains [70,73,74]. Electroporation has been used in the transformation of *Anabaena* sp., *Synechococcus* sp., *Synechocystis* sp., *Fremylla diplosiphon* and *Plectonema boryanum*, [70,73,74]. Ultrasonic transformation has been employed in the transformation of *Synechocystis* sp. [70,73,74]. It was noted that, in *Synechocystis* 6803, exogenous DNA is integrated into the genome following natural transformation by means of a well-established double homologous recombination system [72,73]. Cyanobacteria that use systems with homologous recombination mechanisms allow for more effective gene modifications to wild-type genes [9]. This is true since gene replacement depends on homologous recombination in order to replace a section of the recipient chromosome with a donor DNA that contains a selectable marker [75]. Williams and Szalay [76], were the first to demonstrate homologous recombination in cyanobacteria as a means of transformation, while Golden and Wiest, [77] were the first to achieve homologous recombination in *Anabaena*. Although the transformation of cyanobacteria was thoroughly investigated, very little is known on the mechanism by which DNA binds to the cell, gets processed and subsequently transported across the membrane during transformation [10,72,78]. However, it is thought that type IV pili are involved in the uptake process [79,86]. Studying the effect of deleting the gene(s) encoding type IV pili may give an insight into their role in the transformation process. There is no doubt that the development of tools for the modification of cyanobacteria will help accelerate cyanobacterial research which will ultimately improve its use for industrial biotechnology applications. However, such improvements should not be restricted to the highly investigated unicellular model organisms but extended to include strains with desired characteristics such as those with fast growth rates, resistance to high temperatures and salinity, or resistance to infection and nitrogen fixers to mention a few. This will increase the spectrum of cyanobacteria used for industrial application purposes where characteristics such as the ones mentioned above are crucial.

### 3.2. Selectable Markers

It is essential to be able to select for rare transformant events amongst a large population of treated cyanobacterial cells [4,11]. Systematic and efficient methods whereby successful transformants can be identified and isolated are well established. Usually, host cells that contain the transferred DNA are identified by transferring a gene whose product has the potential to inactivate an antimetabolite, such as an antibiotic [22]. When cultured in the presence of the antimetabolite, only those cells to which the gene (normally an antibiotic resistance cassette) has been transferred should grow. Since cyanobacteria are Gram-negative bacteria, antibiotics and antibiotic-resistant genes that are known to be effective for that group of bacteria have been extensively exploited [11].

Antibiotics used as selectable markers in cyanobacteria, in general, include neomycin and kanamycin and the corresponding neomycin phosphotransferase (npt)-encoding genes from transposons Tn5 and Tn903, and streptomycin and spectinomycin and the corresponding aminoglycoside adenyltransferase (aadA)-encoding genes from Tn7 [87] and the omega interposon [88]. Bleomycin, although expensive, is sufficiently active to be cost effective with *Anabaena* 7120. Chloramphenicol and chloramphenicol acetyl transferase (cat) have proven useful for unicellular cyanobacteria [89] and *Anabaena* sp. strain 90 [90]. Nourseotricine has been reported recently to be effective in a study on an RNA polymerase mutant of *Synechocystis* 6803 [91]. Antibiotics such as tetracycline and rifampicin, which are light sensitive, are of much reduced usefulness for cyanobacteria [21,22]. Other antibiotics with the same property include erythromycin [10], zeocin [92] and gentamicin [93]. However, gentamicin and erythromycin use in cyanobacteria induces oxidative damage and should be used with caution [94,95].

*Synechocystis* 6803 specifically shows preference to some antibiotics. For example, it has been reported that the transformants grew 4–6 days earlier in the plates containing kanamycin than those containing chloromycetin [73]. Furthermore, the bacterial promoter and termination sequences of kanamycin is recognised by *Synechocystis* 6803 and hence can be used without further modifications [11]. A summary of the selectable markers (antibiotic resistance markers) suitable for use in *Synechocystis* 6803 can be found in (Table 2) while the same combinations of antibiotics can be tested with closely related species of cyanobacteria. By using a combination of antibiotic resistance cassettes, multiple genes can be inactivated in a single cyanobacterial strain [96,97,98]. However, it should be noted, when using multiple antibiotics resistance cassettes, that in some cases using one type of antibiotic resistance cassette may lead to resistance towards another. For example, the cassette normally used for spectinomycin resistance also results in streptomycin resistance. Therefore, it is necessary to introduce the streptomycin resistance cassette before the spectinomycin resistance cassette [11].

### 3.3. Plasmid Expression Vectors

Plasmid vectors that are used in transformation of cyanobacteria can be classified into two main groups [22]. The first group is the non-replicating plasmids or integrative plasmids and these cannot replicate independently and would eventually be lost through successive cell divisions [26]. The other group is commonly referred to as replicative plasmids, shuttle vectors or biphasic plasmids and are capable of replication in both cyanobacteria and *E. coli* [22,26,37] as they most often contain replication sequences from both *E. coli* and the cyanobacterial host [21]. Both integrative and replicative plasmids have been developed for cyanobacteria and have their experimental uses. For example, the ideal plasmid vector for transporting DNA that must either transpose (e.g., a transposon) or integrate (e.g., by homologous recombination) in order to be stably maintained are the integrative vectors [6,86]. These vectors are ideal for targeted mutagenesis in which the gene in the chromosome is replaced by the mutated gene in the plasmid [6] or in which the cyanobacterial host needs to select for the recombination of the plasmid into the chromosome [76]. It has been observed that none of the common cloning vectors from *E. coli* seem to be able to replicate in cyanobacteria [4], presumably because the colE1 origin of replication, carried on vectors such as pUC19, pBluescript and pGEM-T, are not recognised by the DNA-replication machinery of cyanobacteria [86].

Replicative plasmids can be classified into two types: those that contain replicons of broad-host range plasmids [102,103] and those derived from endogenous cryptic plasmids [37,104]. Replicative plasmids carrying a variety of selectable markers have been constructed for cyanobacteria [6]. Examples include the pFCLV 7 [89] and plasmids derived from RSF1010 (e.g., pSL1211, pPMQAK1 and pFC1) [26,29,102]. Replicative plasmids that can replicate in multiple hosts (either broad or narrow host range) have the potential to express an exogenous gene in numerous hosts [22]. Nevertheless, most shuttle vectors for cyanobacteria are still not well characterised for their copy number [21], and the lack of shuttle vectors with a varied copy number puts limitations on the controlled expression of heterologously expressed genes by selecting their copy number [99]. For example, the copy number from plasmids derived from RSF1010 are ten per chromosome in *E. coli* cells and approximately 10–30 per cell in *Synechocystis* 6803 (~1–3 per chromosome) [102,103]. In a recent study, it was reported that the GT wild-type strain contains 142 genome copies in the exponential phase and 42 genome copies in the linear and stationary growth phase [105]. The copy number of other broad host-range plasmids have not been quantified to date [29], which adds a limitation to their use. A recent review on shuttle vectors used in various cyanobacterial hosts is presented in [26].

In terms of transformation efficiency, it has been reported that transformation of *Synechocystis* 6803 with plasmids obtained from *Synechocystis* 6803 cultures result in 10–20 times better transformation rates than transformations done using bacterial plasmids, perhaps due to a compatible DNA methylation pattern. Nevertheless, plasmids extracted from *E. coli* are routinely used as they are much more convenient to prepare [106].

## 4. Emerging Synthetic Biology Technologies

Several tools for the model cyanobacterium *Synechocystis* 6803 have recently been developed [12,100,101,107,108,109]. The existing methods for distinguishing viable from non-viable cyanobacterial cells are either time-consuming (e.g., plating) or are not easy to prepare (e.g., fluorescent staining). Recently, a cell-viability fluorescence assay was developed that provides a quick and simple way of detecting viable from non-viable cells in a cyanobacterial culture [107]. In this method, red chlorophyll fluorescence and an unspecific green autofluorescence were used for the differentiation of viable and non-viable cells without the need of sample preparation. This method can be used to monitor the fitness of cultures grown for novel industrial biotechnological products [107]. Standardised cloning methods, like shotgun cloning or cDNA cloning [110], have been employed to express exogenous genes or knockout native genes in *Synechocystis* 6803 [111,112,113,114,115,116]. For maximum expression levels of heterologous genes, codon optimisation has been utilised in several studies with favorable effects [16,116,117,118]. Additionally, the development of degradation tags used for the fine-tuning of protein levels and protein turnover times in this model organism have also been achieved [100].

Such advances aid in the generation of more complex metabolic systems that have the potential to oscillate in phase with circadian rhythms of cyanobacteria. Furthermore, recombinant DNA technology has been employed in *Synechocystis* 6803 for the increased production of carotenoids [101]. In this study, the psbAII promoter was used to drive the transcription and overexpression of coding sequence of several genes involved in carotenoid biosynthesis. The genes cloned in *Synechocystis* 6803 were the yeast isopentenyl diphosphate isomerase (*ipi*) and the *Synechocystis* β-carotene hydroxylase (*crtR*) and the linked *Synechocystis* genes coding for phytoene desaturase and phytoene synthase (*crtP* and *crtB*, respectively). The expression of *ipi*, *crtR*, and *crtP* and *crtB* led to a large increase in the corresponding transcript levels that led to the accumulation of zeaxanthin and myxoxanthophyll in the mutant strains [101]. A study involving increased production of carotenoids from *Synechocystis* 6803 reported a β-phellandrene yield reaching 1% of the dry cell weight [116]. This was accomplished by expressing β-phellandrene synthase as a fusion protein with the β-subunit of phycocyanin. The expression of enzymes of the mevalonic acid (MVA) pathway and geranyl-diphosphate synthase (GPPS) increased substrate availability and overall terpenoid biosynthesis pathway utilisation [116]. Another study reported a pinene synthase mutant with an altered cofactor preference towards magnesium, which possess superior activity to the wild-type [119]. Co-expression of a less active farnesyl-diphosphate synthase (FPPS) increased production of pinene by allowing pinene synthase to outcompete FPPS for GPP [119].

A quick way to create knockins and knockouts and introducing a trans-operon in cyanobacteria was developed [108]. In this method, the open reading frames (ORF) for each desired protein are codon optimised in silico for the chosen cyanobacterial chassis. The intergenic region sequences identified in highly expressed native operons are then added to the sequences to create overlapping parts suitable for assembly where the synthesised parts are assembled in one step [108] into the transformation vector, downstream of a regulatable promoter. The resulting plasmid is then used to transform the chassis strain that contains a negative selectable marker (–ve) inserted into a neutral locus. Two homologous recombination events allow replacement of the marker with the trans-operon, with selection based on the loss of the marker, which results in a marker-less transgenic line. The aim for using intergenic operon sequences taken directly from well-understood and highly expressed operons within the host was to avoid internal transcriptional terminator sequences and to prevent any self-hybridisation or secondary structure in mRNA transcripts. Expression of the trans-operon is induced by a transcriptional activator encoded by a “regulator” gene in the chassis strain such that inducer X (e.g., a metal ion) triggers production of the activator (Figure 2).

Since the level and strength of gene expression is key for higher product titers, which is of industrial biotechnology interest, a lot of attention has been directed to find ways to increase expression of foreign genes in cyanobacteria. Current methods for the regulation of gene expression in cyanobacteria include native and heterologous promoters that can be regulated by different inputs [26] and the use of strong inducible promoters. The use of different types of inducible promoters in cyanobacteria has been described in detail [29]. A list of inducible promoters that can be used in cyanobacteria can be found in [29] and thus will not be discussed here. Recently, however, efficient heterologous protein production from cyanobacteria was accomplished by utilizing a newly discovered strong promoter (P_cpc560_) [109]. The promoter consists of two predicted promoters from the cpcB gene (sll1577) and 14 predicted transcription factor binding sites (TFBSs) that are crucial for its promotor strength. It was found that using this promoter to express two genes, encoding enzymes involved in metabolism in *Synechocystis* 6803 resulted in the buildup of functional proteins at a level of up to 15% of total soluble protein in the cyanobacterium. This level is comparable to that obtained using the *E. coli* expression system, which demonstrated the strength and efficiency of P_cpc560_. These findings demonstrated the potential of utilizing P_cpc560_ for the efficient production of recombinant proteins in cyanobacteria.

New methods for the precise regulation of gene expression became also available. For example, theophylline-responsive riboswitches were utilised for the effective regulation of gene expression in four diverse cyanobacterial species [12]. The riboswitches were evaluated for the expression of a yellow fluorescent protein reporter in *Synechococcus elongatus* PCC 7942, *Leptolyngbya* sp. strain BL0902, *Anabaena* sp. strain PCC 7120, and *Synechocystis* sp. strain WHSyn. The study demonstrated that the riboswitches were effective and superior in regulating gene expression than the commonly used isopropyl-β-d-thiogalactopyranoside induction of a lacI^q^-P_trc_ promoter system. It has also been shown that the new system can be used to regulate toxic genes (*sacB*), demonstrating the potential of utilizing riboswitches to regulate proteins that are detrimental to biomass accumulation. Such advancement in the field of synthetic biology and genetic engineering in diverse cyanobacteria facilitate the development of microalgae bioengineering.

Another tool that has a particularly important industrial application is the development of gene replacement methods that allow for the creation of antibiotic-free genetically modified cyanobacteria (marker-less mutants) [101]. Such strains can be created using the toxic levanusucrase gene (*sacB*) [120] that is illustrated in Figure 3. In addition to the *sacB* method, a more recent method of creating marker-less mutants has been developed for and demonstrated in the model cyanobacterium *Synehcocystis*. In this method, a *mazF* expression under the nickel-inducible, *nrsB*, promoter was used [121]. The *mazF* gene, encodes an endoribonuclease that cleaves mRNA at the ACA triplet sequence, inhibiting protein synthesis. To create the marker-less mutants, the *nrsB* promoter controls expression of *mazF* gene and an antibiotic resistance cassette (in this case *aphII* encoding kanamycin resistance was used) is also alongside the *mazF* gene (under the control of its own promoter). The first selection was carried out on kanamycin, and the second selection on nickel. Attempts were later directed towards reducing the number of cloning steps required to make marker-less transformants using the *sacB/nptI* cassette [120]. Another marker-less mutant system for *Synechocystis* 6803 and *Synechoccocus elongatus* PCC7942 was also developed using the FLP/FRT recombination system [122]. This system uses a self-replicable shutter that contains a flipase (FLP) gene from *Saccharomyces cerevisiae*, which is introduced into the strains of *Synechocystis* 6803 and *Synechoccocus* 7942 where the genome has already been transformed with a kanamycin resistance gene and a flipase recombination target (FRT) flanking either side [122].

A novel system utilised for generating marker-less cyanobacterial mutants is CRISPR/Cas9 [123,124]. CRISPR/Cas9 (clustered regularly interspaced short palindromic repeats/CRISPR associated protein 9) was discovered in bacteria and archaea as an adaptive immune response to foreign DNA [125]. Initially, invader DNA is incorporated as a spacer, which is transcribed into RNA strands that direct the Cas9 protein to complimentary invader DNA to be cleaved [126]. By engineering the spacer sequence to complement genetic targets, certain genes can be silenced or replaced with desired sequences. The main advantage of using this method for marker-less mutant generation is the ability to target several points for mutagenesis simultaneously [127,128,129]. There has been some work aimed at characterizing native CRISPR systems in cyanobacteria [130,131,132,133], yet very limited work on CRISPR/Cas9 application. A recent study has demonstrated that Cas9 is toxic to *Synechococcus elongatus* UTEX 2973, yet marker-less mutants have been achieved through transient expression of the Cas9 gene [123]. CRISPR interference (CRISPRi) has also been successful in the suppression of several gene targets in cyanobacteria [124,134]. Although this system has not been applied on cyanobacterial strains for a long time, its implementation for multiplex genome editing has potential of saving time and effort.

## 5. Conclusions and Future Prospective

In the past decade, cyanobacterial research attracted a great deal of attention that has declined recently due to biological restrictions of native strains at industrial-scale production [135]. This has indicated that the use of cyanobacteria as an industrial platform requires the intervention of genetic engineering. Due to advancements of synthetic biology tools and their implementation on cyanobacterial organisms, a wide range of products have been achieved from direct CO_2_ conversion. Most of the current work is focused on understanding cyanobacterial metabolic systems for tailoring strains to be more resilient and productive. Advancements in sequencing technology that reduced cost and increased output resulted in more cyanobacterial genomes being fully sequenced.

“Omic” studies that compare certain cyanobacterial strains with the model ones will shed light on important factors that could accelerate progress of genetic engineering in these organisms. One example is focusing research on discovering the cause behind the inability of certain cyanobacterial strains to naturally transform with exogenous DNA, which could be enabled by incorporating a competence gene, like *comF* of *Synechocystis* 6803 [136], or silencing certain nucleases genes. in silico modeling, utilising flux balance analysis and systems biology, is a promising route for production pathway optimisation of select strains [81]. This is valid for strains with well-annotated genomes, like *Synechocystis* 6803, which makes achieving industrial-scale feasibility possible perhaps in the “near” future.

The novel technique CRIPSR/Cas9 (clustered regularly interspaced short palindromic repeats/CRISPR associated protein 9) has immensely advanced genomic editing of several organisms, yet its implementation on cyanobacteria is still in early stages and more research is required to evaluate the potential of this method. For industrial application, expression levels have to be enhanced for the production process to generate acceptable profitability. Strong promoters, like P_cpc560_, are essential for good expression levels of desired products and thus further research is needed to test such strong promoters on different cyanobacterial strains and production of different products. Introduction of complete pathways for novel products could be possible in selected cyanobacterial strains by careful evaluation and analysis of in silico models, exploitation of the CRISPR/Cas9 editing technique and the use of a strong promoter.

The application of new genetic engineering tools on other cyanobacterial strains will expand the platform of cyanobacterial use in industrial biotechnology. However, it is important to combine genetic tools with other disciplines such as biochemistry in order to gain better understanding of pathway engineering that may lead to novel industrial products or improve titers of existing ones. Therefore, this combination should be done with a solid understanding of the microbial metabolism at the system level in order to comprehend and overcome the effects of product toxicity that result from production and overexpression of novel products. Product excretion and removal from the culture has successfully reduced the toxic effect of certain products [23,137,138]. Nevertheless, research has to be directed towards a reliable method of transporting or excreting the products with minimal effect on the health of the cells. Another idea of bypassing the toxic effect of products on cyanobacterial cultures is by silencing certain stress response genes, specifically the ones responsible for inducing apoptosis [139,140]. Better understanding of cyanobacterial metabolic systems along with development of new genetic engineering tools will help tap into the great potential of cyanobacteria as a multipurpose industrial “cell factory” of the future.

## Figures and Tables

**Figure 1 life-06-00042-f001:**
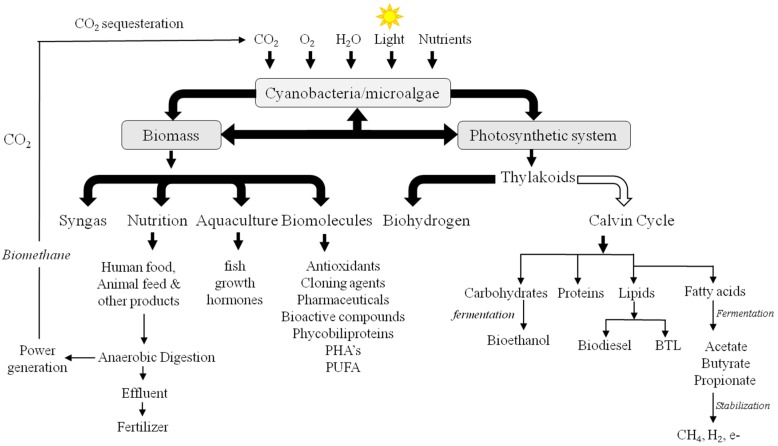
Schematic overview of different biotechnological uses and potential products of genetically engineered cyanobacteria. Items under “Photosynthetic system” refer to native products that are directly linked to the functions within the thylakoid membrane. The photosynthetic system diverges electrons from two primary reactions in the thylakoids directly to H_2_. The products of the Calvin cycle (carbohydrates, proteins, lipids and fatty acids) lead to the formation of various biofuel options. The white arrow indicates products under light conditions, while the black arrow indicates products under dark conditions. Items under “Biomass” refer to uses of the entire biomass and heterogeneous products through genetic engineering. PHA's: Polyhydroxyalkanoates, PUFA: Polyunsaturated fatty acids, BTL: bipolar tetraether lipids. All the arrows indicate the relative utilization or production of an item. The double ended arrow between “Biomass” and “Photosynthetic system” implies that biomass is a result of the photosynthetic system.

**Figure 2 life-06-00042-f002:**
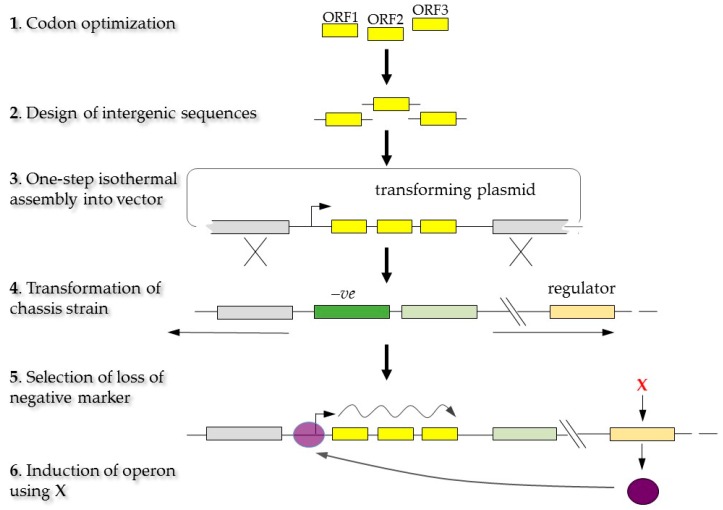
Generation of marker-less mutants through homologous recombination using a codon optimised trans-operon. Yellow boxes are the open reading frame (ORF). The green box is a gene used as a negative marker (-ve). The purple circle is the activator that is produced by addition of inducer X.

**Figure 3 life-06-00042-f003:**
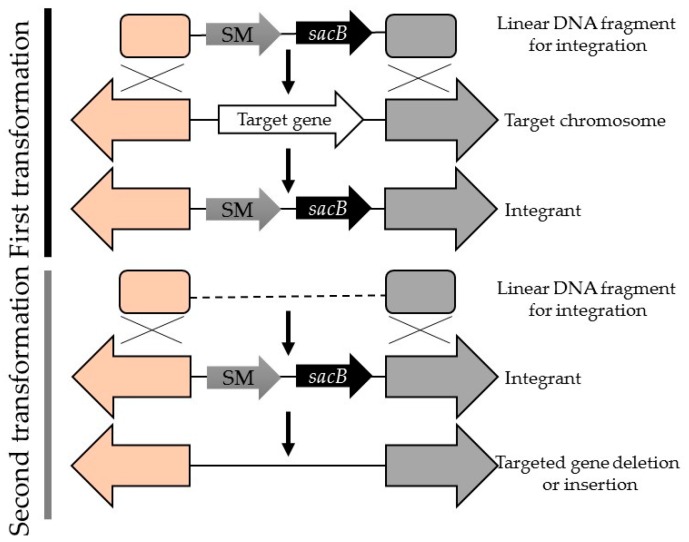
Generating marker-less mutants through consecutive insertion and deletion of *sacB* gene. The colors of the arrows and squares on each end are different to indicate the upstream and downstream regions of the selectable marker (SM) and *sacB* and that certain boxes match with certain arrows for recombination to occur.

**Table 1 life-06-00042-t001:** Current reports of biofuel production in genetically engineered cyanobacteria. The number of transgenes required is indicated, although in several cases additional genes were introduced to increase productivity.

Fuel Molecule	Species Engineered	No. of Genes	Reference
Ethanol (C_2_H_5_OH)	*Synechocystis* sp. PCC6803	2	[20]
	*Synechococcus elongatus* PCC7942	2	[21]
	*Anabaena* sp. PCC7120		Algenol Biofuels Company
1-Butanol (C_4_H_9_OH)	*Synechococcus elongatus* PCC7942	5	[22]
Isobutanol (C_4_H_9_OH)	*Synechococcus elongatus* PCC7942	5	[17]
	*Synechocystis* sp. PCC6803	2	[23]
Ethylene (C_2_H_4_)	*Synechococcus elongatus* PCC7942	1	[8,24]
Isobutyraldehyde (C_4_H_8_O)	*Synechococcus elongatus* PCC7942	5	[17]
Isoprene (C_5_H_8_)	*Synechocystis* sp. PCC6803	1	[16]
Free fatty acids (C_10_–C_18_)	*Synechocystis* sp. PCC6803	5	[25]
Fatty alcohols (C16/C18)	*Synechocystis* sp. PCC6803	1	[26]
*n*-alkanes (CnH2n)	*Synechococcus* sp. PCC7002	2	Joule Unlimited Company
	*Thermosynechococcus elongatus* BP-1		Joule Unlimited Company
	*Synechocystis* sp. PCC6803		Joule Unlimited Company
Hydrogen (H_2_)	*Synechococcus elongatus* PCC7942	1	[27]
Sesquiterpenes (C_15_H_24_)	*Synechocystis* sp. PCC6803		[28]
Squalene	*Synechocystis* sp. PCC6803	Inactivation of gene *slr2089*	[29]

**Table 2 life-06-00042-t002:** Selectable markers (antibiotic resistance markers) suitable for use in *Synechocystis*.

Selectable Marker	Source	Concentration (µg/mL)	References
Chloramphenicol	pBR325	5–150	[9,90]
Erythromycin	pRL425	5–300	[9]
Kanamycin	pUC4K (Tn 903)	5–500	[9,90]
Spectinomycin	pHP45Ω	3–250	[6,9]
Neomycin	Tn5	NA	[32]
Streptomycin	Tn7	NA	[99]
Spectinomycin	pHP45Ω (Tn7)	25	[90,99]
Zeocin	-	25	[100]
Gentamicin	-	NA	[101]

NA: not available.

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
