# Peer review of "Cyanobacteria as Chassis for Industrial Biotechnology: Progress and Prospects"

_life, 2016, doi:10.3390/life6040042_

Round 1
Reviewer 1 Report
The manuscript ””Cyanobacteria as chassis for industrial biotechnology: progress and prospects” by Al-Haj et al. provides a nice addition to recently published review articles about utilization of cyanobacteria in biotechnology. Please find below few suggestions how to further improve the manuscript.
1. Since the aim is in industrial scale biotechnology I suggest that authors include a chapter about available large scale growth options for cyanobacteria.
2. In addition to the mentioned antibiotic resistance cassettes also nourseotricine casette functions well in cyanobacteria (Koskinen et al. 2016 Molecular Microbiology 99:43-54). It should be also noted that some antibiotics including erythromycin and gentamicin induced oxidative stress. The use of these antibiotics might be problematic if the modification effects on ROS scavenging systems, see Cameron and Pakrasi 2011 Applied and Environmental Microbiology 77:3547-3550.
3. What is the difference between white and black arrow in Fig.1? In the current version photosynthesis and biomass are separated, but actually without photosynthesis no biomass is produced. To readers not familiar with cyanobacteria also connections between thylakoid-biohydrogen, thylakoid-Calvin cycle and Calvin cycle related products will be difficult to understand without more detail figure legend.
4. Some paragraphs only describing contents of next couple of paragraphs could be removed (Lines 123-126; 244-249.
5. The reference list contains many typos, for example it is Synechocystis sp. PCC 6803 not synechocystis sp. Pcc6803.
Author Response
The manuscript ”Cyanobacteria as chassis for industrial biotechnology: progress and prospects” by Al-Haj et al. provides a nice addition to recently published review articles about utilization of cyanobacteria in biotechnology. Please find below few suggestions how to further improve the manuscript.
Response: Many thanks to the reviewer for his complements.
1. Since the aim is in industrial scale biotechnology I suggest that authors include a chapter about available large scale growth options for cyanobacteria.
Response: We agree with the reviewer that the topic of large scale growth options for cyanobacteria is an interesting one to discuss. However, this is out of the scope of the current review as the review focuses on the metabolic engineering and synthetic biology tools available for cyanobacteria. Discussing the available large scale growth options for cyanobacteria would be a good topic for a separate publication.
2. In addition to the mentioned antibiotic resistance cassettes also nourseotricine casette functions well in cyanobacteria (Koskinen et al. 2016 Molecular Microbiology 99:43-54). It should be also noted that some antibiotics including erythromycin and gentamicin induced oxidative stress. The use of these antibiotics might be problematic if the modification effects on ROS scavenging systems, see Cameron and Pakrasi 2011 Applied and Environmental Microbiology 77:3547-3550.
Response: Thanks to the reviewer for the suggestions. The mentioned points have been included in the manuscript.( Line: 182-183, Line:186-188)
3. What is the difference between white and black arrow in Fig.1? In the current version photosynthesis and biomass are separated, but actually without photosynthesis no biomass is produced.
Response: The figure has been modified and a more detailed legend has been added. (Line: 73)
To readers not familiar with cyanobacteria also connections between thylakoid-biohydrogen, thylakoid-Calvin cycle and Calvin cycle related products will be difficult to understand without more detail figure legend.
Response: New legend added (Line: 87)
4. Some paragraphs only describing contents of next couple of paragraphs could be removed (Lines 123-126; 244-249).
Response: We agree with the reviewer and removed these paragraphs.
5. The reference list contains many typos, for example it is Synechocystis sp. PCC 6803 not synechocystis sp. Pcc6803.
Response: The references have been formatted correctly and the typos corrected.

Reviewer 2 Report
Comments for authors
The present paper is a review of the field of cyanobacterial industrial biotechnology. The authors summarize some results in the field and proceed to present overviews of ”omics” and genetic engineering techniques as applied to cyanobacteria.
General comments:
The introduction focusses on some examples of products generated in cyanobacteria, but there are many reviews of this field out there by now, and the short, incomplete summary given here is not very useful for the reader looking for an overview of the field. Furthermore, the information selected gives a rather random expression, and, more importantly, the references used are incorrect in many cases. As examples, from page 1 and 2: on line 32, line 36, line 38, line 42, line 48, line 50, line 33, line 60, line 65, line 67, line 68, line 69 references used are either completely wrong or unsuitable as source for the statement they are meant to support. This problem continues throughout the paper (some, but not all, instances are listed under “specific comments” below).
The subsequent sections on techniques have the topics of ‘omics, genetic engineering and synthetic biology. While the authors do cover some of the latest development, the information selected and the treatment of each topic is incomplete and it is difficult to see how the selections were made. As examples: Why not include groundbreaking RNA sequencing results, when microarray data is described? Why describe in detail the use of sacB for counter selection, and why is that in the section of synthetic biology? Why classify viability screening as synthetic biology, but not mention experiments with codon optimized genes, or standardized cloning procedures?
Specific comments
L43-44: It is not appropriate to use the whole title of a paper, verbatim, to describe its contents.
L53-58: Ethylene is neither a fish growth hormone, nor a PHA or a PUFA.
L67-L69: Of the references given for “developing genetic engineering tools...”, no less than 6 are concerning expression of mosquitocidal proteins. The selection of references seems to be just any paper describing transforming of these strains of cyanobacteria for any purpose. Each reference should be selected based on its importance for the statement it supports, in this case the references should describe the development of tools for model strains.
L148-149: repeats statement from above paragraph about natural transformation.
L151-152: What is meant by ”almost all exogenous DNA is integrated”?
L163-164: ”there is no doubt that...” Improving DNA transfer is not at all a bottleneck in genetic engineering. This sentence does not make sense. It is true that extending the techniques to strains currently not possible to transform would be a step forward, but that is another issue.
L196: this is rather trivial.
L196-198: Not necessary if, for example, the antibiotics used are Cm/Km.
L208: ”...according to their mode of replication...” , ”...first group is the non-replicating...” Non-replication is a mode of replication?
L219-222: ”Interestingly, ... ” The fact that certain origins of replication work in certain organisms and not in others is old knowledge and can hardly be described as something of particular interest for cyanobacteria. Furthermore, the reference 90 used is far too recent.
L227, 231, 233: wrong refs 38, 114,111, 112.
L251: Gibson assembly (ref 120) is not a new tool for Synechocystis.
L259: Ref 112 used degradation tags, 2010.
L270-281, including fig 2, ”A quick way...” : This section seems to describe a general work flow for genetic engineering, but is difficult to follow as the purpose is not explained. What is the idea with the highly expressed native operons? What is the negative marker? This is a normal procedure, not anything special. The only reference given is to Gibson assembly, which has nothing to do with what is being described (except for the assembly of DNA constructs, which is not at all specific to the method described here).
L290-291: Again using the title of the paper to describe its results.
L294-295: ”important enzymes”, what enzymes? Why mention they are ”important” if they are not important enough to explain? You could just say ”two genes”.
L305: ”more superior”.
L314: Incorrect references for sacB counter selection.
Whole section L311-341: This description describes in detail what was in the L270-281 section only mention in passing as use of a negative selectable marker, and should probably have come first. However, the topic is thoroughly reviewed many times since at least the early nineties and does not warrant this in depth treatment.
L343, L349L355: Refs 129 and 132 concerns CRISPRi. Throughout the section, the authors seem confused about mutation vs interference. CRISPRi is not a technique for introducing mutations.
L354-355: CRISPRi is well understood.
L368-369: Lack of natural competence would not be a problem if other methods worked. Much work has been done in strains which cannot be naturally transformed (Nostoc PCC7120!) and this is not a bottleneck for engineering.
L393-394: Why would toxic effects be bypassed by silencing stress responses? Also, wrong ref, the reference details stress responses but says nothing about silencing and toxic effects.
Reference section: Most references are wrongly formatted.
Author Response
Comments for authors
The present paper is a review of the field of cyanobacterial industrial biotechnology. The authors summarize some results in the field and proceed to present overviews of ”omics” and genetic engineering techniques as applied to cyanobacteria.
General comments:
The introduction focusses on some examples of products generated in cyanobacteria, but there are many reviews of this field out there by now, and the short, incomplete summary given here is not very useful for the reader looking for an overview of the field.
Response: The focus of this review is the molecular techniques and systems that has been or can be applied on cyanobacteria and not the products in particular.
Furthermore, the information selected gives a rather random expression, and, more importantly, the references used are incorrect in many cases. As examples, from page 1 and 2: on line 32, line 36, line 38, line 42, line 48, line 50, line 33, line 60, line 65, line 67, line 68, line 69 references used are either completely wrong or unsuitable as source for the statement they are meant to support. This problem continues throughout the paper (some, but not all, instances are listed under “specific comments” below).
Response: The references have been corrected.
The subsequent sections on techniques have the topics of ‘omics, genetic engineering and synthetic biology. While the authors do cover some of the latest development, the information selected and the treatment of each topic is incomplete and it is difficult to see how the selections were made. As examples: Why not include groundbreaking RNA sequencing results, when microarray data is described?
Response: More recent examples RNA sequencing application have been added (Line: 93-96, 97-99). There is also a paragraph after the microarray data that presents several examples of RNA sequencing studies in cyanobacteria that are more specific to biofuels.
Why describe in detail the use of sacB for counter selection, and why is that in the section of synthetic biology?
Response: We agree with the reviewer and removed the detailed description of sacB and was mentioned only in lines: 326-329. It is under synthetic biology as it involves the re-design of the cyanobacterial natural system for a useful purpose (generating mark less mutants is important for industrial application purposes).
Why classify viability screening as synthetic biology, but not mention experiments with codon optimized genes, or standardized cloning procedures?
Response: Studies utilizing standard cloning procedures and codon optimization have been included in Line: 255-260.
Specific comments
L43-44: It is not appropriate to use the whole title of a paper, verbatim, to describe its contents.
Response: We agree with the reviewer. It has been changed (Line: 42-44).
L53-58: Ethylene is neither a fish growth hormone, nor a PHA or a PUFA.
Response: Yes, the selection of wording might have led to a confusion. We removed “ For instance” to correct the meaning (Line: 55).
L67-L69: Of the references given for “developing genetic engineering tools...”, no less than 6 are concerning expression of mosquitocidal proteins. The selection of references seems to be just any paper describing transforming of these strains of cyanobacteria for any purpose. Each reference should be selected based on its importance for the statement it supports, in this case the references should describe the development of tools for model strains.
Response: We do not agree with the reviewer here as the statement we wanted to support is that genetic engineering tools have been developed for many model organisms and indeed the references selected describe genetic engineering tools developed for model organisms. However, since there are a few examples on using new tools for the expression of mosquitocidal proteins, we have removed a few of these references.
L148-149: repeats statement from above paragraph about natural transformation.
Response: The repeated sentence has been removed (Line: 146-147).
L151-152: What is meant by ”almost all exogenous DNA is integrated”?
Response: We removed "almost all" to clarify the confusion (Line: 149).
L163-164: ”there is no doubt that...” Improving DNA transfer is not at all a bottleneck in genetic engineering. This sentence does not make sense. It is true that extending the techniques to strains currently not possible to transform would be a step forward, but that is another issue.
Response: The sentences have been rewritten (Line: 161-163).
L196: this is rather trivial.
Response: We disagree with the reviewers comment. Using the correct antibiotic cassette and paying attention to the fact that in some cases the order of introducing the cassette will affect the experiment is not trivial. A lot of time and effort can be lost from not knowing this fact.
L196-198: Not necessary if, for example, the antibiotics used are Cm/Km.
Response: The sentences have been rewritten to clear any confusion (Line: 198-201).
L208: ”...according to their mode of replication...” , ”...first group is the non-replicating...” Non-replication is a mode of replication?
Response: We agree that was confusing, the sentence has been corrected (Line: 211-212).
L219-222: ”Interestingly, ... ” The fact that certain origins of replication work in certain organisms and not in others is old knowledge and can hardly be described as something of particular interest for cyanobacteria. Furthermore, the reference 90 used is far too recent.
Response: The text has been edited (Line:234).
L227, 231, 233: wrong refs 38, 114,111,112.
Response: Ref 38 has been moved to a more appropriate place (Line: 236). Ref 114 has been replaced with a new ref (Line: 234) and references 111 and 112 have been removed from Line:236.
L251: Gibson assembly (ref 120) is not a new tool for Synechocystis.
Response: The word "new" has been removed, yet it is worth mentioning this method as part of the techniques applied on Synechocystis.
L259: Ref 112 used degradation tags, 2010.
Response: Ref 112 has been added to support the statement in Lines: 260-262.
L270-281, including fig 2, ”A quick way...” : This section seems to describe a general work flow for genetic engineering, but is difficult to follow as the purpose is not explained. What is the idea with the highly expressed native operons? What is the negative marker? This is a normal procedure, not anything special. The only reference given is to Gibson assembly, which has nothing to do with what is being described (except for the assembly of DNA constructs, which is not at all specific to the method described here).
Response: Another reference was added describing the method displayed in figure 2 (Line: 283). The Gibson assembly describes the method for the construction of long stretches of DNA to be used for genetic manipulation, which fits in this section. The idea behind using highly expressed native operons has been clarified (Line: 291-294). A specific negative marker is not mentioned here because this method could apply to several negative markers.
L290-291: Again using the title of the paper to describe its results.
Response: Statement has been rephrased (Line:306-307).
L294-295: ”important enzymes”, what enzymes? Why mention they are ”important” if they are not important enough to explain? You could just say ”two genes”.
Response: The word "important" has been removed, yet it is worth mentioning that these enzymes are involved in the metabolic processes of Synechocystis 6803 (Line: 310).
L305: ”more superior”.
Response: "more" has been removed (Line: 320).
L314: Incorrect references for sacB counter selection.
Response: This section has been shortened because sacB has already been described in detail previously.
Whole section L311-341: This description describes in detail what was in the L270-281 section only mention in passing as use of a negative selectable marker, and should probably have come first. However, the topic is thoroughly reviewed many times since at least the early nineties and does not warrant this in depth treatment.
Response: We agree with the reviewer that this method has been reviewed before, so the description of sacB method has been shortened (Line:326-329).
L343, L349L355: Refs 129 and 132 concerns CRISPRi. Throughout the section, the authors seem confused about mutation vs interference. CRISPRi is not a technique for introducing mutations.
Response: We have corrected the cites of this section. However, We would like to direct the reviewer to ref 128, which shows that CRISPR/Cas9 could be used to knockout genes. We have also included 3 more references indicating the possibility of multiplex genome editing by using CRISPR/Cas9 (Line:354-355).
L354-355: CRISPRi is well understood.
Response: The sentence has been rephrased (Line: 360-361).
L368-369: Lack of natural competence would not be a problem if other methods worked. Much work has been done in strains which cannot be naturally transformed (Nostoc PCC7120!) and this is not a bottleneck for engineering.
Response: We agree with the reviewer's comment, but we have not mentioned that lack of natural competence is a problem or that it's a bottleneck for engineering. The message expressed here is that natural competence would reduce work and perhaps time required for genetic engineering of some strains.
L393-394: Why would toxic effects be bypassed by silencing stress responses? Also, wrong ref, the reference details stress responses but says nothing about silencing and toxic effects.
Response: The sentence has been rephrased to better convey the intended message (Line: 398-400).
Reference section: Most references are wrongly formatted.
Response: The references have all been reformatted and the typos corrected.
Reviewer 3 Report
see attached document

Author Response
In this contribution Lamya Al-Haj and colleagues review a number of aspects of applications with cyanobacteria in the synthesis of products from water, CO2 and sunlight. The English grammar used for the text is fine, but with respect to contents the paper reads as if it was completed a few years ago, and was very recently updated with some details on synthetic biology and Crispr/Cas (examples: recent review at bottom of page 6 is from 2012 and ‘recent developments’ in third paragraph of page 7 are from 2009 to 2012).
Response: This review is intended to a broad audience, thus some of the older methods that are already well known (still being used in labs) have been included. Several concerning the citation, several more recent references have been included throughout the manuscript.
Furthermore, the promise at the end of the introduction, i.e. “that insights will be given on commercial applications of cyanobacteria”, is not fulfilled.
Response: The statement has been changed to better represent the data displayed (Line: 69-71), yet examples of commercial applications of cyanobacteria has been also presented throughout the manuscript.
The various chapters are at times very outdated, like the one on natural transformation and the making of marker-less mutants.
Response: Even though these methods are outdated, yet they are still used in the lab. Since this manuscript is intended to for a broad audience, we decided to keep them for a broader overview. Several new references have also been added in different chapters to update the content.
The paragraph on synthetic biology misses key papers on application of metabolomics and proteomics techniques (i.e. ref 1, 2).
Response: We would like to thank the review for this suggestion. Ref 1 and 2 have been added to the synthetic biology section (Line: 100-107).
For proof of success in synthetic biology of terpene production reference is made to 2.5-fold overproduction of one of the endogenous carotenoids of Synechocystis, while much more impressive work on this aspect has been published by colleagues from USA, China, Japan, etcetera (see e.g. ref 3, 4).
Response: The synthetic biology section has been modified to include more recent studies (Line: 272-281).
Also, Fig 2 does not present the most straightforward approach in making marker-less mutants and is complemented by an incomplete legend.
Response: Figure 2 is a simple illustration of one of the approaches to make markerless mutants, which seems acceptable to reviewer 4.
As the field reviewed in this contribution is filled with recent reviews already, and mostly with papers of much better quality (e.g. ref 5, 6 and ref 7 from most of the current authors), I cannot recommend publication of this contribution. References: 1] Pade N, Erdmann S, Enke H, Dethloff F, Dühring U, Georg J, Wambutt J, Kopka J, Hess WR, Zimmermann R, Kramer D, Hagemann M. Insights into isoprene production using the cyanobacterium Synechocystis sp. PCC 6803. Biotechnol Biofuels. 2016. 2] Borirak O, de Koning LJ, van der Woude AD, Hoefsloot HC, Dekker HL, Roseboom W, de Koster CG, Hellingwerf KJ. Quantitative proteomics analysis of an ethanol- and a lactate-producing mutant strain of Synechocystis sp. PCC6803. Biotechnol Biofuels. 2015 Aug 5;8:111. 3] Formighieri C, Melis A. Sustainable heterologous production of terpene hydrocarbons in cyanobacteria. Photosynth Res. 2016 Feb 19. [Epub ahead of print] 4] Tashiro M, Kiyota H, Kawai-Noma S, Saito K, Ikeuchi M, Iijima Y, Umeno D. Bacterial Production of Pinene by a Laboratory-Evolved Pinene-Synthase. ACS Synth Biol. 2016 Jun 21. [Epub ahead of print]. 5] Case AE, Atsumi S. Cyanobacterial chemical production. J Biotechnol. 2016 Aug 10;231:106-14. doi: 10.1016/j.jbiotec.2016.05.023. 6] Gao X, Sun T, Pei G, Chen L, Zhang W. Cyanobacterial chassis engineering for enhancing production of biofuels and chemicals. Appl Microbiol Biotechnol. 2016 Apr;100(8):3401-13. 7] Gomaa MA, Al-Haj L, Abed RM. Metabolic engineering of cyanobacteria and microalgae for enhanced production of biofuels and high-value products. J Appl Microbiol. 2016 Jul 13. doi: 10.1111/jam.13232. [Epub ahead of print] Review.

Reviewer 4 Report
This paper is a review of the use of cyanobacteria as vehicles for industrial biotechnology. The scope covers traditional products and processes through to aspirational goals such as biofuels. It covers well established techniques including selectable markers and plasmid vectors as well as more recently developed approaches; "omics", bioinformatics (indirectly), synthetic biology and gene editing. I found the approach was very reasonable and the review flowed well. The descriptions were accurate and the material useful for a broad audience, for example to workers wanting to move, from eukaryotic algae to cyanobacteria, or who wish to simply keep abreast of a field which is not their primary area. I would anticipate that readers would be keen to hear more about both "progress" and 'prospects" - for example a somewhat more visionary list of applications employing CRISPR, assuming the preliminary problems of implementation are overcome. What are the larger ambitions of workers in this field, now that such a powerful tool has been (or is likely to be) delivered into their hands?
The text itself is in need of careful and detailed revision. To guide the authors a few examples are provided below, but the entire text needs attention. The writing is flabby, with unnecessary circumlocutions, phrases and repetition as well as awkward constructions and some ambiguities. It contains colloquiallisms which need to be deleted. Terms are sometimes used inconsistently. While the word count could be reduced with no loss of information, more importantly the precision could be improved. If the primary authors are fluent in spoken English but not so attuned to the subtleties of written (scientific) English, the responsibility falls to the other authors to provide the necssary scrutiny to bring the writing up to a professional standard.
A few infelicities are provided as examples:
Line 35: "Recently, a lot of attention has been granted…"
Line 87: "…There is a vast amount…"
Line 87: "…"a lot of microarray data"
Line 84: "…"alkanes-producing and non-producing"
Line 83: "…"biologically-relevant'"
Line 111: "…"saving immense time"
Line 118: "Although, recently…"
Line 137: "an extracellular nucleases"
An example of how the text can be condensed:
Line 105: " Over the past ten years, several models have been made to represent the metabolic pathways of Synechocystis 6803 [76-86]. The main reason for doing so, was to use the information to improve the production of biofuels in cyanobacteria."
Why not say something like "Over the past decade, Synechocystis 6803 metabolism has been modeled [76-86] to understand and improve cyanobacterial biofuel production." Even if the authors feel this loses an essential subtlety in their message I'm sure they could improve on the former sentence.
While the standard of the 3 figures is workmanlike and of reasonable quality (if not exciting) I would welcome additional figures to usefully illustrate the techniques described. In a review, figures do not present experimental data and therefore should help clarify the text.
I believe that with careful revision of the text, this manuscript could be a worthwhile review for its intended audience.
Author Response
This paper is a review of the use of cyanobacteria as vehicles for industrial biotechnology. The scope covers traditional products and processes through to aspirational goals such as biofuels. It covers well established techniques including selectable markers and plasmid vectors as well as more recently developed approaches; "omics", bioinformatics (indirectly), synthetic biology and gene editing. I found the approach was very reasonable and the review flowed well. The descriptions were accurate and the material useful for a broad audience, for example to workers wanting to move, from eukaryotic algae to cyanobacteria, or who wish to simply keep abreast of a field which is not their primary area. I would anticipate that readers would be keen to hear more about both "progress" and 'prospects" - for example a somewhat more visionary list of applications employing CRISPR, assuming the preliminary problems of implementation are overcome. What are the larger ambitions of workers in this field, now that such a powerful tool has been (or is likely to be) delivered into their hands?
Response: Not much work has yet been done on cyanobacteria using CRISPR in terms of industrial application. Yet a few ideas have been mentioned in Line: 458-465.
The text itself is in need of careful and detailed revision. To guide the authors a few examples are provided below, but the entire text needs attention. The writing is flabby, with unnecessary circumlocutions, phrases and repetition as well as awkward constructions and some ambiguities. It contains colloquiallisms which need to be deleted. Terms are sometimes used inconsistently. While the word count could be reduced with no loss of information, more importantly the precision could be improved. If the primary authors are fluent in spoken English but not so attuned to the subtleties of written (scientific) English, the responsibility falls to the other authors to provide the necssary scrutiny to bring the writing up to a professional standard.
A few infelicities are provided as examples:
Line 35: "Recently, a lot of attention has been granted…"
Corrected
Line 87: "…There is a vast amount…"
Corrected
Line 87: "…"a lot of microarray data"
Corrected
Line 84: "…"alkanes-producing and non-producing"
Corrected
Line 83: "…"biologically-relevant'"
Corrected
Line 111: "…"saving immense time"
Corrected
Line 118: "Although, recently…"
Corrected
Line 137: "an extracellular nucleases"
Corrected
An example of how the text can be condensed:
Line 105: " Over the past ten years, several models have been made to represent the metabolic pathways of Synechocystis 6803 [76-86]. The main reason for doing so, was to use the information to improve the production of biofuels in cyanobacteria."
Why not say something like "Over the past decade, Synechocystis 6803 metabolism has been modeled [76-86] to understand and improve cyanobacterial biofuel production." Even if the authors feel this loses an essential subtlety in their message I'm sure they could improve on the former sentence.
Thank you for the above suggestion. The text was revised to shorten unnecessary or lengthy sentences.
While the standard of the 3 figures is workmanlike and of reasonable quality (if not exciting) I would welcome additional figures to usefully illustrate the techniques described. In a review, figures do not present experimental data and therefore should help clarify the text.
Thank you for your kind comment. The legend in Line 87 has been modified to clarify the figure for audience not familiar with the content.
I believe that with careful revision of the text, this manuscript could be a worthwhile review for its intended audience.
Round 2
Reviewer 2 Report
This is a revised version of a manuscript previously reviewed. Along with the revised version, the authors have submitted answers to point raised by the reviewers. In these answers, they claim to have made changes that have not actually been made, and to have amended issues which have not been amended.
Case in point: In my previous review, I raised the concern that many references are used in the wrong context or are completely irrelevant for the statements where they are placed in the text. It should not be necessary to point out that this is a serious issue for a review article. To help the authors identify the type of problem I saw, I gave a number of of examples from the manuscript’s first two pages to illustrate my point. The authors now answer to this issue that "The references have been corrected". However, they remain in every instance unchanged in the new version. This gives an impression of carelessness and even disrespect for the work of several reviewers on this manuscript, and it is not the level of work I would expect from the authors of this paper.
More examples where the issues I raised have not been adressed, or have not been fully amended:
In response to my comment on including standardized cloning procedures, the authors have introduced references to “standard cloning”, as a basic molecular biology protocol. Standardized cloning in a synthetic biology context was of course meant to describe methods and standards such as BioBricks, Golden Gate assembly and the like.
In response to my comment that Gibson assembly is not a new tool for Synechocystis, and pointing out that the reference was used wrongly, the authors removed the word “new” but left the references intact so that the text still describes Gibson assembly as a technique for Synechocystis.
Also regarding the Gibson assembly, they completely disregarded my comment that Gibson assembly cannot be described as “A quick way to create knock ins and knock outs and introducing a trans-operon in cyanobacteria” (lines 415-416 in the new version). They only claim in their answer that they have added another reference (they have not, at least not in the section discussed), and that Gibson assembly “fits in this section”. That may or may not be so. The problem was that the reference is inappropriate where they have placed it, since the topic they write about there is NOT Gibson assembly.
There are many more instances of remaining problems and inadequate answers, but I cannot list and explain them all one more time here.
Reviewer 3 Report
A large part of my initial criticism to this paper still stands: “the paper reads as if it was completed a few years ago, and was very recently updated with some details on synthetic biology and Crispr/Cas”. On many small aspects a proper response has been given, with the corresponding changes in the manuscript, but also in several noticeable respects this was not done, like for instance the procedure for making marker-less knock-outs (see ref 1). Also the problem of genetic segregation – of key importance in view of the polyploidy of many cyanobacteria – is not addressed.
In this revised version nothing has been done to fulfil the promise of the title: “Cyanobacteria as chassis for industrial biotechnology”, because details on metabolic modeling, metabolomics assays, and the efficiency of photosynthesis, are completely lacking.
So I can only agree to acceptance of this contribution if its title is changed from: “Cyanobacteria as chassis for industrial biotechnology: Progress and Prospects” to: “Cyanobacteria as chassis for industrial biotechnology: A review of recent transcriptomics and proteomics studies and molecular genetic engineering”.
Ref 1: Cheah YE, Albers SC, Peebles CA (2013) A novel counter-selection method for markerless genetic modification in Synechocystis sp. PCC 6803. Biotechnol Prog. 29(1): 23-30.